# Assessing the Acceptability of Home-Based HPV Self-Sampling: A Qualitative Study on Cervical Cancer Screening Conducted in Reunion Island Prior to the RESISTE Trial

**DOI:** 10.3390/cancers14061380

**Published:** 2022-03-08

**Authors:** Dolorès Pourette, Amber Cripps, Margaux Guerrien, Caroline Desprès, Eric Opigez, Marc Bardou, Alexandre Dumont

**Affiliations:** 1IRD, Ceped (IRD, Université de Paris, Inserm), 75006 Paris, France; eric.opigez@ird.fr (E.O.); alexandre.dumont@ird.fr (A.D.); 2Ethno Logik, 97435 Saint Gilles les Hauts, La Reunion, France; ambercripps@gmail.com; 3AgroParisTech, 75005 Paris, France; margaux.guerrien@agroparistech.fr; 4Centre de Recherche des Cordeliers, Sorbonne Université, Université de Paris, Inserm, Laboratoire ETREs, 75006 Paris, France; caroline.despres@bbox.fr; 5CIC1432, CHU Dijon-Bourgogne et Université de Bourgogne, 21000 Dijon, France; marc.bardou@u-bourgogne.fr

**Keywords:** cervical cancer screening, home-based self-sampling, HPV testing, Reunion Island, socio-economic insecurity, social disadvantage, social isolation, qualitative study

## Abstract

**Simple Summary:**

Reunion Island is a French overseas department where cervical cancer is 2 to 3 times more prevalent than in mainland France. Screening rates are lower, especially among women from disadvantaged socioeconomic backgrounds. The RESISTE research program aims to assess whether sending a self-sampling kit to women’s homes could improve screening uptake. Prior to the implementation of this program, qualitative research was conducted with women from various disadvantaged backgrounds, as well as health professionals, to measure knowledge levels, identify barriers and triggers to screening, and assess the acceptability of a home-based self-sampling methodology. The results demonstrate the interest of women and health professionals in this screening method, while also highlighting the need to (1) reassure women regarding usage and quality and (2) provide support and outreach.

**Abstract:**

Cervical cancer incidence and mortality rates are 2 to 3 times higher in the overseas department of Reunion compared with mainland France. RESISTE’s cluster-randomized controlled trial aims to test the effectiveness of home-based self-sampling (HBSS) through a high-risk oncogenic papillomavirus test sent out by post to women who have not been screened in the past 3 years, despite having been invited to do so through a reminder letter. Prior to the trial, qualitative research was carried out to understand screening barriers and assess anticipated acceptability. Semi-structured interviews were conducted with 35 women and 20 healthcare providers. Providers consider HBSS a viable method in reaching women who tend not to visit a doctor regularly, or who are reluctant to undergo a smear pap, as well as those who are geographically isolated. They considered, however, that women would require support, and that outreach was necessary to ensure more socially isolated women participate. The majority of the women surveyed were in favour of HBSS. However, two-thirds voiced concerns regarding the test’s efficiency and their ability to perform the test correctly, without harming themselves. Based on these findings, recommendations were formulated to reassure women on usage and quality, and to help reach socially isolated women.

## 1. Introduction

Potentially curable and preventable, cervical cancer (CC) is the third most common gynaecological cancer before the age of 45 [1]. The Global Cancer observatory, hosted by the International Agency for Research on Cancer, estimates that there were 604,127 cases and 341,931 deaths from cervical cancer worldwide in 2020 [2]. In 2020, cervical cancer ranked as the ninth most common cancer worldwide, and the fourth in women [2]. It is estimated that, in France, 21% of cervical cancer cases are attributable to social disadvantage [3]. Of those affected by cervical cancer, women living in socio-economically deprived conditions are more likely to die from the disease [4]. 

Reunion Island, a French overseas department, is one of the poorest departments in France (after Mayotte and Guyana). The standardized mortality rate for cervical cancer is 4.8/100,000 women, two times higher than in Metropolitan France [5]. Since 2009, as part of the Organized Cervical Cancer Screening (OCCS) program, an invitation is sent out by post to women who have not undergone CC screening in the past 3 years [6]. The participation rate of women who receive such invitations remains low (23–24%) [7]. A 2017 survey of a representative sample of 1000 women showed the main determinants of non-adherence to OCCS to be lower socioeconomic status, a lack of knowledge about CC screening, low literacy rates, and immigration from other Indian Ocean territories [7]. 

Since 2020, cervical cancer screening in France has evolved from pap-smear to clinician-collected HPV testing. Self-sampling for HPV testing is reported to be a good alternative to clinician-collected samples, promising increased participation in cervical cancer screening [8]. It offers significant benefits over conventional sampling in terms of cost, coverage, and convenience for patients. Self-sampling reaches high-risk groups who currently have limited access to national health system screening for personal and practical reasons [9]. Several studies comparing self-sampling and clinician-collected samples for HPV detection show good concordance when clinically validated PCR-based HPV tests are used [10,11]. However, if vaginal self-sampling (VSS) has been suggested as an alternative to clinician-collected cervical sampling for HPV testing, in unscreened or under-screened women, it has not turned into clinical practice yet. A good proportion of women (63%) interviewed in the 2017 survey were nonetheless in favour of home-based self-sampling (HBSS) CC screening [7]. 

The RESISTE project is an effectiveness and implementation hybrid trial. It is incorporated into the organized cervical cancer screening system of four departments in France, including Reunion Island. The objective of this trial is to increase CC screening among women residing in socio-economically disadvantaged areas by sending out a HPV self-sampling kit with the reminder letter [12]. 

Before launching the intervention, RESISTE conducted an informative qualitative study. The primary objectives were to better understand the reasons for the low adherence to OCCS in Reunion Island and to assess the acceptability of the RESISTE intervention from the perspective of both women and health providers. The secondary objectives of this study were to highlight disadvantaged women’s understanding, perceptions, practices, and experiences surrounding cervical cancer and screening. We thus aimed to better identify internal and external barriers as well as triggers towards screening, including through a home-based self-sampling methodology.

## 2. Materials and Methods

This qualitative study is based on semi-structured interviews conducted with 35 women and 20 health providers in Reunion Island between November 2020 and April 2021. 

### 2.1. Site Description Section

Reunion Island is a French overseas department located in the southwest of the Indian Ocean. In 2018, 39% of households in Reunion were living below the national poverty threshold, compared with 15% in mainland France (Grangé 2021). Households in rural and disadvantaged urban areas are those most subject to poverty, whereas the west and the north of the island are generally less economically disadvantaged [13].

Being a former French colony, the population is composed of people who arrived on the island during colonial times and are of African, Indian, or French descent, and people who have arrived more recently and are of European origin (mostly French) or from other Indian Ocean islands (mostly Mayotte—another French Overseas Department, Madagascar, the Comoros, and, less often, Mauritius).

Previous qualitative studies have emphasized that Reunion Island is characterized by medical pluralism, linked to the history of the island and its multi-ethnic character [14]; this is particularly the case in the field of cancer [15,16]. Research on maternal and perinatal care has shown that women remain attached to ancestral practices, often used in parallel with more conventional medical care or screening methods [17]. Practices vary according to women’s social and ethnic backgrounds. Furthermore, several publications suggest that women living in Reunion Island have imperfect control over their sexuality, and more generally over the prevention of associated risks, such as unwanted pregnancies or sexually transmitted diseases [18,19].

### 2.2. Interview Guidelines

The aim of the interviews conducted with women was to gather experiences in terms of cervical cancer screening, to identify obstacles and levers to accessing screening, and to assess acceptability of a home-based self-sampling methodology, while taking into account women’s backgrounds and various other constraints encountered in managing their health [20]. Based on literature on the subject and our previous work [15,21,22,23,24,25], we developed interview guidelines addressing different themes: knowledge and perceptions of women’s illnesses, in particular of cancer and CC; understanding surrounding prevention and treatment; knowledge, practices, and experiences relating to screening; triggers and barriers to screening; women’s relationship to providers; practices around intimate health; and appreciation of an HBSS methodology. The aim of the interviews with health professionals was to document their practices in terms of CC screening and identify any difficulties encountered in screening disadvantaged women, while also weighing their opinion and knowledge of HPV self-sampling (see interview guidelines in Appendix A and Appendix B).

### 2.3. Target Populations

Respondents were selected so as to include women from a wide variety of backgrounds and, in particular, those from disadvantaged backgrounds.

The smallest administrative division in France is called an IRIS (Ilots Regroupés pour l’Information Statistique/Grouped Islands for Statistical Information), corresponding to a population of around 2000. In order to identify the areas of residence of women from disadvantaged backgrounds, we used a social deprivation index to target more socio-economically deprived areas of the island displaying a higher index of socio-economic insecurity. This index is designed from four groups of variables available in Reunion: employment/socio-professional category, housing, education/training, and marital/family/household status. We chose to conduct the study in the most disadvantaged IRIS, classified in the 4th and 5th quintile (see Figure 1). In these areas, we then contacted local town halls and social workers, maternal and child protection centres (MCPC), family planning centres (FPC), communal social action centres (CSAC), local charities and organisations, and a prison. We asked these structures to put us in contact with women of various profiles identified as being in vulnerable/disadvantaged situations, and with health professionals working with such women.

Figure 1 below shows the geographical distribution of the 35 women and 20 health professionals interviewed by socio-economic insecurity index, based on the social deprivation index:

We selected a diverse sample of women in terms of age, area of residence and remoteness, origin and cultural background, literacy levels and education, occupation, and poverty status. Women in particularly disadvantaged situations or at potentially higher risk were also included: prison inmates [26], sex workers [27], the homeless [28], the disabled [29], and women with an addiction [30]. Care was taken to include women over 50 years of age, in order to understand the reasons why this age group in particular tends to miss CC screening. We also selected a diverse sample of health professionals in terms of profession (doctors, midwives, gynaecologists, and pharmacists), health facility (public and private), target population, and location (more or less remote). We believe sampling has enabled us to achieve a representative view of the diverse situations experienced by women in relation to CC screening. We stopped recruiting women when we reached data saturation, i.e., when the interviews no longer provided us with new data/information or new results [31].

### 2.4. Interview Conduct and Analysis

Prior to participating in the interviews, all participants were informed of the purpose of the study and its non-mandatory and confidential nature. They were provided with an information sheet and an informed consent form to sign. In order to ensure confidentiality was respected, women’s names were neither mentioned nor recorded at any point during the interviews. 

Semi-structured interviews were conducted by an anthropologist, a member of the research team (AC), with the support of a research assistant (MG). Interview guidelines were developed for each target group (women and professionals). The interviews with health professionals were conducted in French. The interviews with women were conducted in French, Creole (local to Reunion Island), and Shimaore (local to Mayotte), depending on women’s preferences, and with the help of an interpreter when necessary. All interviews were recorded with the consent of the participants and then transcribed and translated into French when necessary. We then conducted cross-sectional thematic analysis of the transcribed interviews: responses to each theme discussed during the interview were first identified and then compared to those of other participants. The themes corresponded to the main lines of research enquiry and were based on the interview guidelines (see above). General trends and divergences were identified and analysed [32] by contrasting the responses based on women’s age, location, cultural background, screening history (regular or irregular, up-to-date or not), past screening experience (obstetrical violence), factors of past or present vulnerability (multiple sexual partners, sex workers, a history of sexual violence), and social disadvantage/isolation (remoteness, imprisonment, housing issues—homeless or living in sheltered housing, disability, education levels, language barriers—written and spoken literacy in French).

### 2.5. Ethics

The RESISTE protocol was first approved on 6 February 2020 by the Ethics committee “Sud-Ouest et Outre-Mer II”, protocol version 01 du 9 December 2019—n° ID-RCB: 2020-A0002237 (Dossier 2-20-006 id6698) 2°HPs.

## 3. Results

### 3.1. Study Participants

As presented in Figure 1, participants came from across the island, including more remote areas where access to healthcare is more complicated, such as in “les hauts” (areas at a higher altitude) and in the three natural “circuses” (craters of ancient extinct volcanoes, surrounded by natural ramparts; these remote areas are inhabited, and two are accessible by car, whereas the third is only accessible on foot). 

The study sample was comprised of 35 women of various ages (10 aged 25 to 35, 17 aged 36 to 50, and 8 over 50), and divers social and economic backgrounds, cultural origins and life contexts, as encountered in Reunion Island (see Table 1).

The participants born in Reunion represented the diverse ethnic backgrounds found on the island, as exemplified in Figure 2.

This study found that migrant women (except the 2 from mainland France), women with housing issues, and those in prison had multiplied factors of social disadvantage and isolation. Among the 13 migrant women (excluding the 2 from mainland France), 6 experienced difficulty expressing themselves in French, and 3 said they were illiterate. Of the 6 women experiencing housing issues, 5 had been subjected to family isolation, 3 had experienced GBV, 3 were migrant women, of which 2 faced language barriers, 1 said she had a mental illness, and 1 said she was an alcoholic. Of the 3 women interviewed in prison, 2 had difficulty expressing themselves in French, 1 was a migrant, and 1 said she had been an alcoholic.

Amongst the 20 health professionals interviewed, we were able to cover diverse professions, genders, practice structures, and locations (refer to Table 2 and Figure 1).

### 3.2. Screening Practices

As demonstrated in Figure 3, very few of the 35 women interviewed were found to be up-to-date and have a regular screening history in accordance with the recommended schedule (8 out of the 32 for whom screening is recommended). 

The majority of those who were not up to date with their screening (9 out of 12) had undergone at least one smear test in their lives. Among the three women who had never had a smear test, two were unaware of the existence of the disease and of screening availability; all three women had difficulty speaking French and were unable to read in French. 

The interviews with three women, although not currently concerned by CC screening (a virgin and two women who had undergone a hysterectomy but had a regular screening history prior to the intervention), were nevertheless deemed relevant and included in this study. 

Below we explore some of the factors that lead to irregular screening histories, or a lack of screening, as well as those factors that trigger women to screen. Where meaningful, barriers and triggers have been explored in relation to socio-economic and cultural background and other relevant factors.

### 3.3. Disease Knowledge Levels and Attitudes towards the Disease

#### 3.3.1. Disease Awareness

Although the vast majority of women were aware of the disease’s existence, all those encountered considered their knowledge on disease origins, prevention, development, and treatment, as well as screening practices and recommendations, to be minimal and insufficient. It is worth noting that two women had never heard of cervical cancer, and four others said they had only vaguely heard about the disease (two were from Madagascar, one from Mayotte, and one from Reunion). The two women who had never heard of cervical cancer—a Malagasy woman and a woman from Reunion—displayed factors of social isolation (a woman in sheltered accommodation and another in prison) and language barriers (both had difficulty understanding French and one was illiterate). Both had, however, recently been in contact with healthcare services (the Malagasy woman had given birth to two children in Reunion, and the other woman had received medical screening upon arrival at the prison). 

We found knowledge levels on CC to be highest amongst women who had a relative or close friend who had had HPV or CC (four women), and women who had thought they might themselves have the disease (three women). A further 11 women had come across information on the illness as a result of their general interest in health issues. The remaining 17 had never actively sought information on CC: 11 women said reproductive health was not a topic of conversation with family or friends, 3 said they did not feel concerned by the illness as there was no family history of the disease, and 1 had avoided looking up the illness to “avoid worrying” over it.

#### 3.3.2. Disease Origin Awareness 

When asked how women contract CC, 13 women said they did not know. Interestingly, lack of knowledge was not found to relate to education levels: five of these women had finished their schooling. Amongst the 22 women who provided an answer, 18 women referred to an increased risk amongst women with multiple sexual partners or to STIs. This understanding appeared to be based on the fact that CC affects the genital apparatus, rather than on a precise understanding of how the disease is contracted. Only five women specifically referred to the papillomavirus. 

The second most stated factor leading to CC was genetic predisposition, or a family history of CC (stated by, respectively, five and eight women). Women who believed CC was linked to sexual activity and SDIs appeared to be more likely to screen (12 of these 18 women had a regular screening history) than those who believe CC to be “hereditary” or based on genetics (all 13 women had an irregular screening history). 

#### 3.3.3. Disease Prevention Awareness

Screening and the use of condoms were the most commonly provided means of prevention (each option was stated by 8 women). When questioned about a vaccine, 11 women had heard of some form of immunization (all were from Reunion or mainland France). Women were particularly keen to receive further information on the topic of prevention.

#### 3.3.4. Perceptions of the Disease and of Vital Prognosis

Women considered CC to be, like other cancers, a “serious disease, seen as “terrifying” and “death-like”. Several women (9) qualified the illness as “scary”, with an uncertain vital prognosis. CC was seen as incurable by a third of the women interviewed (12) who considered that “cancer comes back” or “spreads”, even when it appears to be cured, and that treatment can prologue life expectancy but not cure the disease. CC was considered to be curable if detected sufficiently early by 12 women. This study found more pessimistic views on disease prognosis to be related to personal experience: 7 of the 12 women who believed CC to be incurable said relatives/friends who had had cancer had experienced a relapse or generalisation of the disease. More optimistic views were related to higher knowledge levels: 6 of the 12 women who said CC was curable referred to the different stages of development of the disease. 

#### 3.3.5. Known Treatment Options

Treatment options for the disease were perceived to be “heavy duty” and “difficult to bear.” Hysterectomy (stated by 7 women) and chemotherapy (stated by a further 7) were the treatments the most referred to by the 13 women who were able to provide treatment options for the disease. The understanding, exposed by 2 women, that hysterectomy endangers the continuation of a normal sex life can only add to the perceived gravity of the known treatment options.

### 3.4. Screening Barriers and Triggers According to Women and Practitioners

Although the importance of early detection was recognized by 14 women (to “avoid the worst”), their knowledge regarding screening was limited. The act of screening appeared to be dissociated from its purpose.

Regarding **screening knowledge**, Figure 4 shows that just under two-thirds of the women (21) knew the smear is conducted to screen for CC. However, 5 of these women took the purpose of a smear test to also be to detect STDs, urinary infections, “microbes”, and “germs”. This portrays confusion between the smear and the vaginal swab. Such confusion is compounded by the fact that women had undergone smear tests when (and sometimes only when) they were faced with a gynaecological issue (this was the case for 5 women). The other 14 women were unsure both as to how CC was detected and as to the purpose of a smear: they considered the purpose of a smear to be that of a vaginal swab (to detect STIs), or to be part of a general gynaecological check-up conducted to assess “if one is in good health”.

Understanding of the purpose of a smear test did not appear to be directly related to screening regularity. Of 6 women who said they had never sought to know if they had CC, 3 had never had a smear test, while the other 3 had. In fact, 2 women had a regular screening history but had never taken an interest in the disease and had conducted the smear tests out of habit and obligation when they visited a gynaecologist for contraception. Neither woman knew how CC is detected.

**Knowledge of the screening schedule** did, however, appear to impact screening regularity. A total of 12 women did not know that the smear was to be repeated at regular interviews (4 of whom thought the test was only to be conducted once); all 12 women had an irregular screening history. 

Amongst the other 23 women, the frequency the most suggested was every 2 year, a frequency suggested by 12 women, of which 7 had followed the recommended screening schedule. A further 9 women suggested screening needed to be conducted every 6 months to a year—of these 9 women, only 1 had an irregular screening history. The final answer provided by 2 women was “every 5 years”—1 woman was up to date, and 1 was not.

A limited understanding of screening, regardless of prior screening, appears to point towards **a lack of information sharing by physicians conducting smear tests**. This was confirmed by the women interviewed: only 4 felt they had received sufficient information during recent screenings, whereas 17 felt that they had not been adequately informed about the purpose of undergoing a smear test. Some of these women (4) said they “dared not” ask the practitioner for further information. Some practitioners said they avoided sharing “too much” information so as not to scar women, or repeat information they considered women must already be aware of. Practitioners talk systematically about SDIs with young women; they tended, however, to only discuss the screening procedure with those over 35.

While a lack of knowledge on how CC is detected, and on the purpose of smears, can present a barrier to screening, Figure 4 above demonstrates that this was not always the case amongst the women interviewed. Conversely, knowledge regarding CC screening was not always sufficient to ensure a regular screening history. Interviews with women and health care providers helped identify several other factors seen to facilitate or limit access to screening as summarized in Table 3 below.

**Women’s vision of the disease**, of treatment options, and of vital prognosis was found to have an impact on screening behaviour: 8 women said their fear of CC had motivated them to screen for the disease, and a further 8 expressed a “desire to know”, as they considered “prevention better than treatment”. They also considered that they themselves could not “see inside” their own bodies to know if they were healthy or not. They emphasised the need to see a specialist. Although the majority of women said they would prefer to know if they had the disease (20), 2 said they would prefer not to know, as they considered CC to be incurable. 

The professionals interviewed considered fear of the disease a major obstacle to screening and stressed the importance of reassuring women on early detection prognosis and simplified treatment, both at the time of screening and on reception of positive results.

As with knowledge-seeking behaviours, **feeling concerned by the disease** was found to be a factor that encouraged women to screen. Having symptoms suggestive of an STI, having multiple sexual partners, or having a family history of CC were identified by women as likely causes of the disease. These factors were also highlighted by women as triggers to consult a physician, and, in some cases, to specifically request a smear test. A desire to preserve one’s health, or a sense of duty to do so for one’s family, also made women feel concerned by CC and motivated them to screen (7 women). On the contrary, not feeling concerned by the disease appeared to reduce the likelihood of screening: “feeling healthy” was found to be a barrier for 4 women, who did not see the need to screen in the absence of symptoms. 

Age was found to exert an influence on whether women felt concerned by the disease. Young women tended to feel less concerned by the illness (unless they had multiple partners). Receiving an invitation letter stating the recommended screening age was found to counter this barrier and contribute to a sense of urgency to conduct a smear test. Post-menopausal women tended to feel less concerned by screening. 

**The way women experienced screening** shaped the likelihood of regular screening. The context of the consultation generated a certain amount of embarrassment, shame, or exasperation amongst the women encountered. The smear test was felt, by 10 women, to be an invasion of their intimacy: 6 said it “wasn’t easy to be touched”, or indeed that they were “fed up” with “being touched” by physicians, while 4 said they needed to prepare themselves mentally beforehand; a further 2 had been victim to gynaecological violence. Although women said they were able to “overcome” embarrassment, **the fear of pain was found to be an important barrier** to screening. Six women said they were reluctant to conduct smear tests, as they had previously experienced pain during gynaecological examinations; none of these women were up-to-date with their screening. Two said they had experienced obstetrical violence during their first smear test conducted at a young age and had not had another smear test since, or not for 25 years after experiencing “such trauma”. The need to “be gentle” was stressed. 

The logistics of screening and the accessibility of care represent a hindrance for some women who have found it difficult to get an appointment with their gynaecologist, or have been unable to free up time given their work schedule and/or childcare activities (10 women referred to the logistics of attending a medical appointment as either a barrier or facilitating factor).

**Gynaecological consultations undertaken for an unrelated reason** also represent an important lever: cervical screening is integrated into follow-up appointments related to pregnancy, contraception, or the desire to conceive. Nearly half the women interviewed (16) said they had had a smear test during an unrelated gynaecological consultation: 9 when they had had a gynaecological problem or were concerned about STDs, 7 when they were pregnant, and 4 in relation to their contraceptive follow-up. We found that some women no longer underwent screening after gynaecological follow-up had ceased around the age of menopause.

Twelve women said they had themselves requested a smear test either as part of a health routine (5) or when they were concerned about having CC since they had multiple partners (3), they had just become aware of the disease (4) or of the recommended screening age (2), they felt “something was wrong” (3), or a close relative had had the disease (1). A further 2 requested a smear test when they were worried about having a STI (thus confirming low understanding levels of the purpose of a smear).

Most of the women encountered during this study said they paid particular attention to **reminders from health professionals and to screening invitations received through the post** and said they were more likely to undergo a smear test after receiving such reminders. Of the 16 women who had received an invitation/reminder through the post, 11 said the letter had motivated them to subsequently screen for CC—4 of these women had not heard of the disease prior to receiving the screening invitation. Twelve women screened after being encouraged to do so by their general practitioner (the majority waited to see a “specialist” to screen—see below). Furthermore, women who had not received a screening invitation said they would have liked to have been reminded either by letter (4 women) or by their general practitioner (4 women).

Some women perceived screening as an “obligation” imposed by their physician or by the organizations whose logos are displayed on the reminder letter (this point was also stressed by practitioners). This sense of obligation was generally found to contribute positively towards women’s motivation to be screened: 10 women said screening was experienced as a responsibility they take seriously and wish to respect. 

This study found that reminder letters can act as a trigger even for women who are illiterate and/or have difficulty communicating in French (3 women had the letter translated/explained by a family member or physician and subsequently requested screening, 1 illiterate woman did, however, throw such letters away without opening them). The letter is also a means to increase awareness about the disease; 3 women had not heard of CC prior to receiving a screening invitation, all subsequently requested a smear test. 

Some barriers appeared, however, to be difficult to overcome through a letter: 3 of the 4 women who had received an invitation letter and did not subsequently screen said they had no intention of doing so, having had a painful experience in the past.

Finally, women underlined **the role of trust and feeling at ease with the physician** as an important factor contributing to their willingness to undergo screening. For gynaecological examinations, especially Pap smears, women prefer to see a “specialist”: only 5 conducted their last smear test with a general practitioner, while the majority (24) had seen a gynaecologist. Knowing the practitioner beforehand and feeling comfortable to express themselves freely without being judged were described as factors of trust. **The gender of the practitioner** was also found to be decisive in several cases: 13 women said they prefer to see a female practitioner. Four of these women said they would not undergo a smear test with a male practitioner. Although these women preferred a female practitioner, 12 others said they preferred to see a male practitioner, 5 of whom would not undergo a smear test with a female practitioner. Interestingly, the criteria used to justify preferences relating to the practitioner’s gender remained the same irrespective of the actual preferred gender: in both cases, women referred to gender-specific perceptions of professionalism and respect for intimacy.

### 3.5. The Impact of Social Disadvantage and Cultural Factors on Screening 

This study found screening habits to vary according to various internal and external factors that can be influenced by, but are not limited to, social disadvantage and country of origin (see Figure 5 based on data for the 32 women eligible for CC screening). In terms of country of origin, a higher number of women amongst those born in Reunion were found to be up-to-date with screening (13 of 17 women concerned by screening) than amongst migrant women (7 of 13 women). 

Country of origin is also linked to forms of social disadvantage, in particular **language barriers**: of 9 women with limited capacities in spoken and written French, 7 had immigrated from other Indian Ocean territories. Screening was found to be lower amongst migrant women who expressed difficulties communicating in French (only 3 out of 7 were up-to-date) than amongst those who could communicate in French (4 out of 6 were up-to-date). Although language barriers did not appear to prevent awareness of the disease or of the purpose of a smear test, they were found to be a significant barrier to awareness of the screening schedule, and thus to regular screening. Indeed, while half of the 8 women experiencing language barriers knew of the disease and how it is detected, only 1 was aware of the screening schedule; this was also the only woman with a regular screening history. It is noteworthy that language barriers were not found only amongst women who had recently immigrated to Reunion: of the 7 migrant women who had difficulty speaking and reading French, only 1 had recently migrated to Reunion (3 had immigrated more than 5 years ago, and another 3 over 15 years ago).

We also found screening behaviour to be influenced by what we have called ***self-declared ethnicity*** to take into account the cultural impact of women’s place of origin, parents’ cultural origins, and women’s current cultural context and cultural/religious affiliation. Self-declared ethnicity appears to be a determining factor in terms of **communication on reproductive health within families**. Of a total of 15 women who said sex education was not part of their upbringing, we count 2 women from Comoros, 2 women from Madagascar (and a 3rd woman from Reunion with a Malagasy father), all 3 women from Reunion with a Tamil background, and 4 women from more remote areas of Reunion (all 4 said all topics related to intimacy were “taboo”). Religion in itself is not, however, necessarily a determining factor: 4 of the 9 Muslim women said sexuality and reproductive health were openly discussed at home, whereas 5 said the topic was taboo.

**Preferences regarding the practitioner’s gender** were also found to vary according to self-declared ethnicity: Tamil women from Reunion (3), and those from mainland France (2) all preferred to see a woman, while Muslim women (9) and women having arrived from other Indian ocean territories (10) all preferred to see a male practitioner. Women from Reunion, other than Tamil, did not to have a strong preference regarding practitioner’s gender.

In terms of **socio-economic factors,** we did not find economic factors to be a determining factor (all those on welfare benefits had access to free healthcare). We did, however, find women who lived in more extreme situations of poverty and social exclusion to be less likely to be up to date with screening. As presented in the Study Participants section, homeless women and those in prison tended to experience multiple factors of social disadvantage and isolation, which can represent a barrier to screening. Of the 6 women experiencing housing difficulties and the 3 detained women, only 1 had a regular screening history. Both of the women who were unaware of the existence of CC were amongst these women. We also found these women to be less likely to be invited to screen: all were unable to receive post, and 3 said they had not been invited to screen or informed of the screening schedule.

In terms of **geographical barriers**, we found women from the most remote study area, the Cirque of Mafate, accessible only on foot, to have experienced fewer logistical barriers to screening then those from Grand Ilet, a remote village in a circus accessible by car. A helicopter pick-up and drop-off service was in place in Mafate, whereas women from Grand Ilet had to drive over an hour to consult with a gynaecologist, and they also found it difficult to get an appointment.

### 3.6. Attitudes towards Home-Based Self-Sampling

The professionals interviewed were unanimous in recognizing that sending a self-sampling kit to women’s homes should make it possible to reach women whom they are unable to screen: women who tend not to consult a healthcare provider or who are reluctant to have a pap smear, as well as more isolated or disadvantaged women.

A majority of the women encountered said that they would undergo self-sampling (18 out of 31 who answered this question). Some even expressed a certain relief, and were appreciative of the ease of not having to attend an appointment, as well as of the privacy that home-based self-sampling (HBSS) would provide. Others referred to previous self-sampling procedures they had undertaken, stating that they felt at ease with the methodology.

Eight women said they would not self-sample, as they regularly saw a specialist whom they considered would conduct a more thorough examination. Five others were more ambivalent and outlined conditions regarding usage, including being accompanied.

If we look specifically at those women who would be targeted by self-sampling RESIST trail, 9 of the 10 women who were not currently up to date with their screening said they would be happy to perform self-sampling. The remaining woman, who had had a regular screening history until recently, said she would prefer to book an appointment rather than self-sample. 

Although the majority of interviewees said they would self-sample, 23 out of 31 women said they had concerns regarding a HBSS procedure. Their concerns related mostly to usage: 17 women questioned their ability to understand the instructions and perform the sampling procedure correctly, and without harming themselves. Secondly, 7 women expressed concerns regarding the quality of a home-collected sample. They were uneasy about the level of hygiene required to take such a sample, as well as the quality and thoroughness of the procedure as compared with an examination performed at a clinic, and conservation conditions after sampling, given the hot climate. With these concerns in mind, women were uncertain as to whether they would be able to fully trust the results of HBSS CC screening.

The women and professionals interviewed stressed the need for women to be reassured regarding their ability to perform self-sampling and the quality of this screening format. Some professionals related a reluctance to touch one’s intimate parts amongst women, in particular those less educated. Professionals emphasized the need to train and involve physicians in accompanying women through the self-sampling process, so that they can inform them and answer their questions regarding CC and the various available screening methods. Providing a remote support service via a toll-free phone number or on-site support at the level of a practitioner prior to HBSS were some of the ideas put forward by women and practitioners alike. One woman said she would personally prefer on-site self-sampling, as she felt the practitioner would be able to better guide her, be able to check sampling quality, and the sample would be better stored and therefore provide better results.

Several professionals stressed that in order to reach more socially isolated women, self-sampling should be integrated into community outreach activities targeting such populations, and run by organisation having already built up a relationship of trust with targeted populations.

## 4. Discussion

Our results confirm those of prior studies that show women’s level of knowledge regarding CC and screening methods to be low, particularly in disadvantaged populations, and that link low levels of knowledge to lower screening rates [33,34,35]. Lower levels of screening amongst migrant women appeared to correlate with language barriers rather than recent migration. Lower levels of screening amongst migrant women were further found to correlate with lower knowledge of the screening schedule rather than of CC in general. More socially isolated women (women with language barriers, housing issues, or in prison) were found to accumulate factors of disadvantage and be those least likely to screen. They were also found to be less likely to receive an invitation to screen. 

All participants generally felt they had not been adequately informed about CC, prevention, and treatment options. The study results emphasise a poor understanding of the purpose of pap smear testing, even among women who are regularly screened. This highlights a lack, or inadequacy, of communication between health professionals and women, partially attributable to the hierarchical relationship between women and health professionals, but also to communication issues [16]. Previous studies conducted in Reunion have also noted major communication (and cultural) gaps between patients and providers, many of whom are from mainland France. The providers did not speak Creole (and even less so Shimaore or Malagasy), and as this study exemplifies, not all their patients are fluent in French, or, indeed, literate. This accentuates the provider/ patient gap and has been known to even lead to misdiagnosis [16]. 

In terms of cancer-related awareness, few women distinguished between precancerous lesions and cancer; confusion or lack of distinction between the disease and precancerous lesions has also been highlighted in other contexts [25]. This study found greater awareness of the stages of development of the disease to be linked to more regular screening. Higher levels of awareness of the disease, of how it develops, and of the screening schedule were generally found to be strong screening triggers. 

Furthermore, this study found personal experience to play an important role in shaping how the disease and screening are envisioned. These perceptions, in turn, impact women’s screening behaviour. A prior painful experience was found to be a particularly strong barrier to screening. Our findings on the impact of gynaecological violence on screening behaviour are in line with those of other studies conducted among homeless women [21,36], which show a subsequent fear of and refusal to undergo gynaecological examinations. Fear of the disease and fear of “knowing” were also found to represent obstacles to screening as has been found in contexts where disease is socially disqualifying and represents a social and financial burden for the family [25]. In our study, we found fear to be a barrier to screening, specifically when linked to a perception of the disease as incurable and/or leading to a hysterectomy—an intervention perceived as implying a cessation of sexual activity. However, fear of the disease, especially when associated with feeling “at risk” (through sexual behaviour, the presence of symptoms, or a family history of CC) was found to also be a motivating factor, creating a desire to know “before it’s too late”. 

Reminders, in the form of a letter or advice from a general practitioner, were described by women as major external triggers. Interestingly, even those with limited levels of French literacy described the letter as a trigger. 

An equally important motivating factor is the relationship of trust with the healthcare provider. As found elsewhere, gynaecological exams can be perceived as an ordeal for women and an attack on their intimacy [37]. Women sought practitioners perceived to exemplify professionalism and respect for intimacy. Interestingly, some women sought female practitioners for these reasons, whereas others sought male practitioners for the very same reasons. 

With regard to HBSS, health professionals and women were generally supportive of this method of screening, seen as a means of reaching women who are not up to date with their screening. Results show that receiving an invitation to screen through the post can be a strong trigger to greater awareness around CC and the screening schedule, feeling concerned and subsequently requesting a pap smear. The self-sampling kit was considered, in addition, to be more practical and less invasive of one’s intimacy. In a study from Mexico, 96.8% of the participants reported they felt confident carrying out the self-collection themselves [38]. In our study, women did, however, express concern regarding usage and quality. These doubts may be linked to a lack of knowledge of the body and reluctance or difficulty in touching it. This has been found to be the case for women whose bodies have been subjected to deprivation, addiction, and sometimes, violence [21]. Since fear of pain was found to be a major barrier to screening, it is important that women are reassured regarding their ability to perform HBSS without hurting themselves. 

A discrepancy between the acceptability of self-sampling and a lack of confidence to perform it has been described in prior studies on the acceptability of CC self-sampling among women living with HIV in Côte d’Ivoire, Kenya and Rwanda [39,40,41].

## 5. Limitations

This study was conducted in a French overseas department with a specific social, economic, and cultural environment; this may limit the transposability of study results.

Furthermore, as France has just switched to high-risk HPV screening, women and health professionals may not yet have had time to become familiar with this screening modality. Although the women interviewed said they were in favour of this method of screening, this does not necessarily mean they will undertake self-screening. For this reason, a second phase of the study will be conducted post-trial with women who have received the self-sampling kit.

## 6. Conclusions

In Reunion Island, sending out a self-sampling kit for HBSS CC screening appears to be a viable means of reaching additional women who have not had a pap smear for more than 3 years. Discourses of both health professionals and women expressed good theoretic acceptance of the method. A HBSS kit should prove to be, much as a reminder letter is, a means of raising awareness, found to be a trigger to screening. HBSS should help to reduce additional barriers to screening by reducing not only logistical barriers, but also the emotional charge associated with consulting for a pap smear. However, for HBSS to be a success, the kit needs to be accompanied by information aimed at reassuring women on their ability to use the kit, as well as the quality of a self-sampling methodology. Furthermore, although integrating the kit into the standard reminder schedule can provide an acceptable, more convenient and less intrusive alternative to a pap smear, if women with the most irregular screening histories, who are also the most socially isolated, are to be reached, additional outreach measures need to be taken. 

Based on the barriers and triggers to CC screening discussed by the participants in this study, as well as their specific concerns relating to HBSS, we recommend the following for the kit and cover letter:-Clearly and simply explain the importance of early diagnosis, the availability of non-invasive treatment and reassure regarding prognosis;-Present a simplified screening schedule;-Provide other screening options;-Reassure women on usage through simple instructions;-Reassure women on quality (results and sample preservation);-Train local health professionals so that they can better support women throughout the self-sampling process;-Provide accompaniment options: where to ask for advice in person and a toll-free phone number;-Provide additional support for women from categories the most at risk or more socially isolated: work with community organisms to reach women with language barriers and those with housing issues; work through the prison staff to reach incarcerated women;-Integrate HBSS into community health programs targeting socially isolated women.

## Figures and Tables

**Figure 1 cancers-14-01380-f001:**
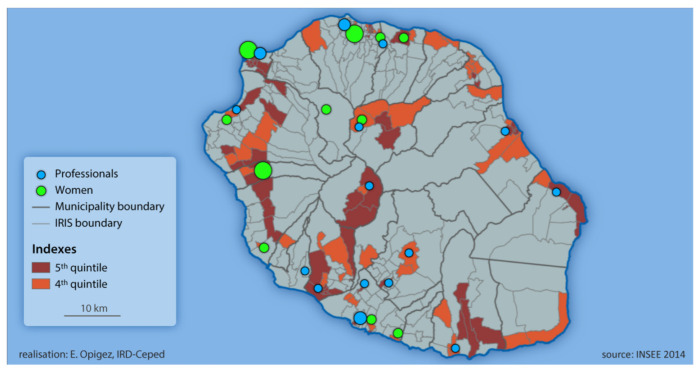
Geographic distribution of study participants.

**Figure 2 cancers-14-01380-f002:**
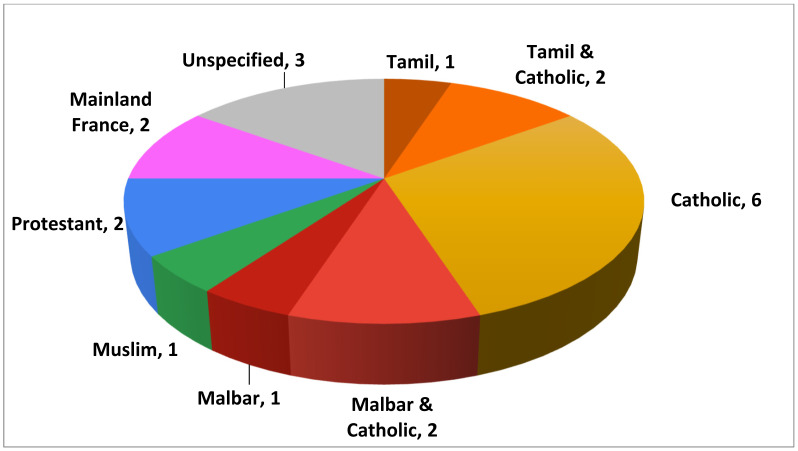
Self-Reported Ethnicity of Women Participants Born in Reunion.

**Figure 3 cancers-14-01380-f003:**
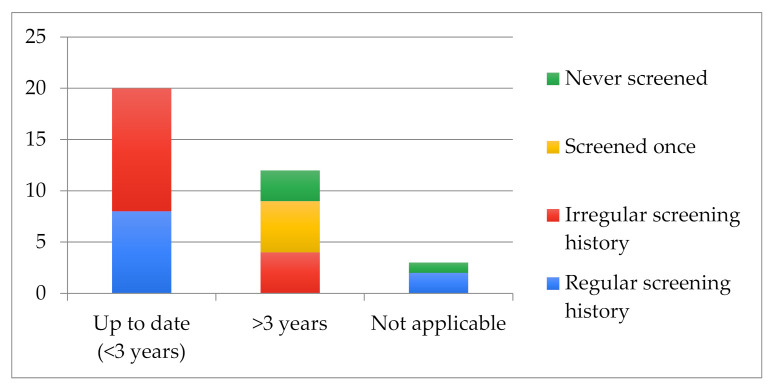
Women’s Screening History by Latest Smear Test Date.

**Figure 4 cancers-14-01380-f004:**
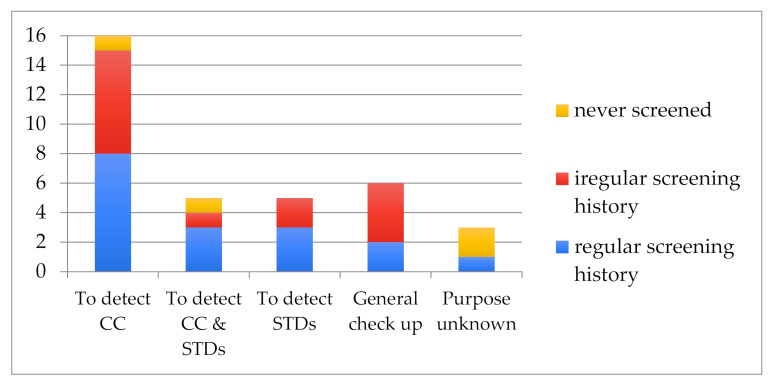
Smear Test Purpose Understanding by screening history.

**Figure 5 cancers-14-01380-f005:**
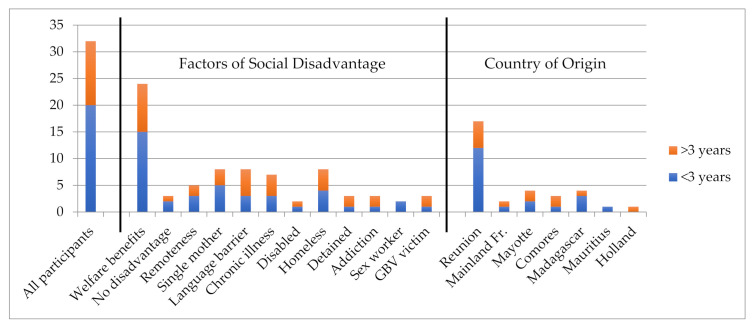
Last Screening Date by Social Disadvantage and Country of Origin.

**Table 1 cancers-14-01380-t001:** Women Interviewees’ Characteristics.

Interviewee Characteristics												
**Place of birth**	Reunion	Mayotte		Comoros		Madagascar		Mauritius		Mainland France		Holland
	20	4		3		4		1		2		1
**Socio-economic criteria**	Unemployed		Employed		Retired		Welfare benefits		Single mother		Remote area	
	24		6		5		24		8		5	
**Factors of sever social disadvantage and isolation**	Migrant		Language Barriers		Illiterate		Homeless		Sheltered Accommodation		In Prison	
	15		8		3		3		3		3	
**Other Criteria**	Family exclusion		Disability		Chronic Illness		Gender Based Violence		Sex Worker		Alcoholism	
	6		3		8		3		2		3	

**Table 2 cancers-14-01380-t002:** Health Professional Participants.

Profession	Men	Women	Details
Medical Doctors	4	2	1 prison doctor and 5 general practitioners: 2 at MCPCs, 1 at a screening unit, 2 private sector practitioners
Gynaecologists	2	2	2 obstetricians working in hospitals, 1 gynaecologist–sexologist working in a screening centre, 1 private sector
Midwives	1	5	1 at a MCPC, 5 private sector practitioners
Pharmacists	2		2 private sector practitioners
Laboratory Practitioners	1	1	1 biologist at a sampling centre, 1 pathologist at an analysis centre

**Table 3 cancers-14-01380-t003:** Summary of Key Barriers and Triggers to Adopting Recommended Screening Behaviour.

Factors of Influence		Barriers	Triggers/Encouraging Factors
Knowledge	CC	Not knowing about CC Believing CC is incurable Believing the treatments offered are difficult to bear Language barriers	Becoming aware of the disease Believing CC is curable if detected sufficiently early Understanding treatment options Being informed by a practitioner
	Screening	Screening schedule unknown	
Attitudes	CC	Not feeling concerned by CC An invisible disease	Feeling concerned by CC
	Screening	Not feeling at ease with the practitioner or the act itself An additional burden in terms of time and obligations Fear of the results	Feeling at ease with the practitioner and the act itself Part of taking care of oneself Not being able to “see inside” oneself Wanting to know
Practices and past experience	Screening	No longer receiving an invitation letter A negative screening experience	Reminders in person or by letter Gynaecological monitoring (contraception, pregnancy…) Integrated into a health routine

## Data Availability

The data presented in this study are available on request from the corresponding author.

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
