# Peer review of "Assessing the Acceptability of Home-Based HPV Self-Sampling: A Qualitative Study on Cervical Cancer Screening Conducted in Reunion Island Prior to the RESISTE Trial"

_cancers, 2022, doi:10.3390/cancers14061380_

Round 1

Reviewer 1 Report

Pourette et al. conducted semi-directive interviews with 35 women from various backgrounds and 20 healthcare providers. The health professionals surveyed consider sending a self-sampling kit to women's homes a viable method in reaching those they are unable to screen. The majority of the women surveyed were in favour of using a self-sampling test at home. However, two-thirds voiced concerns regarding the test's efficiency and their ability to perform the test correctly, without harming themselves. Based on these findings, recommendations were formulated regarding the inclusion of information to reassure women concerning usage and quality.

The claims are properly placed in the context of the previous literature. The experimental data support the claims. The manuscript is written clearly enough that most of it is understandable to non-specialists. The authors have provided adequate proof for their claims, without overselling them. The authors have treated the previous literature fairly. The paper offers enough details of methodology so that the experiments could be reproduced.

Comments

HPV self-sampling has been widely supported by the scientific community following a strong body of literature on the subject. Self-sampling is important in cervical cancer screening as it has been shown to improve participation.

Page 2, line 79, Introduction, add:

“Self-sampling for HPV-testing is reported to be a good alternative to clinician-collected samples, being a promising approach to increase women’s participation in cervical cancer screening (Sultana 2015). It offers significant benefits over conventional sampling in terms of cost, coverage and convenience for patients. Self-sampling reaches high-risk groups who currently have limited access to national health system screening for personal and practical reasons (Mariño 2015). Several studies comparing self-sampling and clinician-collected samples for HPV detection show good concordance when clinically validated PCR-based HPV-tests are used (Petignat 2007, Schmeink 2011). In a study from Mexico, 96.8% of the participants reported they felt confident carrying out the self-collection themselves, and 88.8% reported no discomfort at all performing the procedure (Aranda Flores 2021).”

References

Sultana F, Mullins R, English DR, Simpson JA, Drennan KT, Heley S, et al. Women's experience with home-based self-sampling for human papillomavirus testing. BMC Cancer. 2015;15(1):849. https://doi.org/10.1186/s12885-015-1804-x.

Mariño H, Serra E, Gutiérrez A. Self-sampling is as much effective as gynecologist samples for HPV detection. Medicina Balear. 2015;30:16–20.

Petignat P, Faltin DL, Bruchim I, Tramer MR, Franco EL, Coutlee F. Are self-collected samples comparable to physician-collected cervical specimens for human papillomavirus DNA testing? A systematic review and meta-analysis. Gynecol Oncol. 2007;105(2):530–5. https://doi.org/10.1016/j.ygyno.2007.01.023.

Schmeink CE, Bekkers RL, Massuger LF, Melchers WJ. The potential role of self-sampling for high-risk human papillomavirus detection in cervical cancer screening. Rev Med Virol. 2011;21(3):139–53. https://doi.org/10.1002/rmv.686.

Aranda Flores, C.E., Gomez Gutierrez, G., Ortiz Leon, J.M. et al. Self-collected versus clinician-collected cervical samples for the detection of HPV infections by 14-type DNA and 7-type mRNA tests. BMC Infect Dis 21, 504 (2021). https://doi.org/10.1186/s12879-021-06189-2

Reviewer 2 Report

The manuscript describes a qualitative assessment of cervical cancer screening awareness and attitudes toward HPV self-sampling by women and healthcare providers. This is an innovative approach to understanding barriers to screening in cultural context, and the authors clearly made an effort to be inclusive and thorough in data collection. 

However, there are significant issues that need to be resolved, particularly with regard to the analysis of the interview data (see below for details). Other general comments: the introduction contains a lot of information that is not directly relevant to the study, and not enough background on self-sampling initiatives in other LMICs, the conclusion includes two or three assumptions that do not reflect the results as they are presented, and overall the manuscript could be more focused and concise. Finally, the manuscript would be benefit from English-language editing to resolve grammatical and stylistic issues. Detailed comments are below, I hope these are helpful to the authors.

SPECIFIC COMMENTS:

“Qualitative” seems a more appropriate descriptor than “anthropological”, as the study did not involve immersive fieldwork or ethnography, only qualitative interviews.

Line 29: first invitation to what sort of screening? Provider-collected HPV testing? Cytology?

Line 45: potentially preventable, not curable (?)

Line 52-53: fragment, re-write sentence

Lines 44-57: France mention is out of place in a discussion regarding developing countries, somewhat repetitive –  the same points can be made in a clearer, more concise manner.

Line 61: These are the guidelines for France? Needs a reference

Lines 57-79: this long discussion regarding France is unnecessary, as the project is about Reunion. One or two lines establishing the relationship and inequities between France and Reunion, if at all, is all that is needed and/or relevant.

Lines 80-88: this information should go in the methods as a Site Description section, and needs references regarding poverty rates and breakdown of ethnic groups (by percentage from an official census, for example).

Lines 89-90: What sort of invitation? By mail? Telephone?

Lines 109-136: This needs to go earlier in the introduction

Line: 144-148: Why mention this if it was not used? Irrelevant, can be deleted

Line: what does “anonymously put us in contact mean”? Needs more explanation.

Line 183-184: “new situations”? Rather “new data/information”?

Lines 188-189: confidential, not anonymous – cannot be both (as women gave their consent, signature and personal details, it is not anonymous)

Line 199: need more information regarding the coding process. Was it a single coder or several? If several, was there high reliability between them? Was a coding scheme developed first, and then used by others to thematically analyze the interviews?

It would also be useful to provide a sample of the interview guides that were used, even as an appendix.

Line 213: Why over-represented? 23% of women interviewed were over 50, is that very different from the general population?

Line 217: cirques? What does this mean?

Lines 224-227: were these categories found in interviews? For example, did women speak about their addictions or mental illnesses?

Lines 209-247: is there a way to summarize this information on a table? Some of the detail is not needed, a table with broad categories of participants and providers would be easier to read and provide only the relevant information to the reader

-Line 250-258: same as above

Lines 259-400: I strongly suggest that the authors rework the results section. In the methods, it is mentioned that coding was conducted and that the coding scheme was used to conduct thematic analysis. Neither of these methods are reflected in the results. What codes were used? What were the main themes that emerged?

Usually, themes are reported in terms of frequency: as in, 25% of women mention that “_____” or 70% of women referred to “______”. It is not enough to say “most women felt”, or “some women said”, or “the majority thought”. The most frequent themes are interpreted as the “main” ones – here, some perceptions or beliefs are associated with 1 or 2 individuals, and it is not clear why they are reported as “themes” at all, when they are purely anecdotal.

Also, while the use of direct quotes is a great device to provide context, quotes are not analytic devices. Picking quotes from different respondents without linking them to main themes is not rigorous and seems random cherry-picking.

The Discussion section will change after the results are re-analyzed: it should summarize the findings, but not necessarily repeat every single one (the current discussion could be more concise). It is also inappropriate to bring up new reasons for lack of adherence to CC screening that are not supported by the results. For example, the issue of language barriers or lack of communication between patients and healthcare workers does not come up in the interviews (or at least is not mentioned in the results), yet it is discussed as a barrier in the discussion (Lines 409-418). Lines 437-442 also mention new issues not explicitly explained in the results. Other aspects of the results (such as fatalism, or not wanting to know if the person is sick) are not mentioned at all.

Lines 450-451: again, not supported by the results, so no reason to assume this is the case.

Recommendations should be numbered/bulleted for clarity.

Lines 518-525: this is confusing – is this a new or upcoming study? These new findings need to be explained more thoroughly.

Editing for grammar and clarity is necessary throughout. For example:

-Line 93: “and birth in Indian Ocean islands other than Reunion”

- Line 104: “This can be partly explained…” should be deleted (it’s a generalization that applies equally well anywhere in the world

-Lines 105-107: should go earlier next to the first mention of the survey, then explain that despite this willingness, barriers remain, etc.

-Line 109: unclear

-Line 142: Respondents were selected to include women in…

-Lines 209-212: needs editing, punctuation, etc.

-Lines 218-223: keep tenses consistent

-Lines 232-233: “health professionals who work 232 with women from more socially deprived backgrounds” – no need to keep repeating this. There are several times in the manuscript were descriptors are unnecessarily repeated (as in, “from Reunion islands”, “socially deprived”, etc). It’s already clear who the women and the providers are and where they are located.

Reviewer 3 Report

  • In the manuscript titled “Assessing the Acceptability of Home-based HPV 2 Self-sampling: a Cervical Cancer Screening Qualitative Study 3 conducted in Reunion Island (part of the RESISTE program)”, the authors Pourette et. al. have presented a study that aims to understand the knowledge of cervical cancer and factors that affect HPV screening amongst women in disadvantaged socio-economical situations by interviewing them.
  • While the authors took into account numerous factors, the study lacks strength in terms of the number of subjects in the studied cohort. 35 subjects is far from what would be considered a good sample size representing the population of women in disadvantaged situations, and it makes this study weak. However, the authors do mention that this study is qualitative and that they stopped conducting interviews after saturation in answers was reached. Hence, this small number could be acceptable for this type of a study, if the authors had followed a methodological approach.
  • The results, namely disease and screening knowledge levels, Screening barriers and triggers, and thoughts of women and professionals about home based testing, is listed as a jumbled text of expressions. For example, in the case of screening barriers, It would greatly help the readers if the authors would have represented these results in the form of a table, with subcategories of the kind of screening barriers and the fraction of women that experienced those. Without tabular or pictorial aids, this text is cumbersome to read and the reader finds it hard to draw any final conclusions from the results.
  • Similar to the above mentioned point, the methods lack a systematic approach. For example, the authors list unrelated social factors together, for example illiteracy, ability to speak French, mental illness, physical disability etc. The reader does not know how to draw conclusions from these unrelated factors.
  • The study fails to contribute any new knowledge to what is already known wrt CC and HPV screening amongst women from diverse socio-economical backgrounds. Considering the lack of any new contribution to the field, combined with the lack of a a systematic methodology, this study fails to qualify as a manuscript that would be of interest to the readership of this journal.

Round 2

Reviewer 2 Report

The authors made an effort to address comments from the original review, but the manuscript still needs work to be ready for publication.

In particular, the results section (Lines 278-639) is a detailed summary of every response to every question, but not an appropriate analysis. I strongly suggest the authors refer to the methods and results sections of published manuscripts that utilize a similar approach for guidance. Here are a couple of examples from different settings:

Moucheraud C, Kawale P, Kafwafwa S, Bastani R, Hoffman RM. "It is big because it's ruining the lives of many people in Malawi": Women's attitudes and beliefs about cervical cancer. Prev Med Rep. 2020;18:101093. Published 2020 Apr 8. doi:10.1016/j.pmedr.2020.101093 [there are a lot of good qualitative studies focused on CC in Malawi)

Dzuba, I. G., Calderón, R., Bliesner, S., Luciani, S., Amado, F., & Jacob, M. (2005). A participatory assessment to identify strategies for improved cervical cancer prevention and treatment in Bolivia. Revista Panamericana de Salud Pública18, 53-63.

Below are some other specific comments:

-Use of written number vs. figures (three vs. 3) needs to be consistent throughout the manuscript (see Simple Summary and Abstract, for example)

-Avoid contractions (it’s, who’d) and keep tenses consistent, particularly in results (all should be past tense)

-Punctuation/grammar need to be reviewed, especially the use of commas. See examples:

Line 21-22: “identify barriers and triggers to screening[,] and assess the acceptability…”

Line 34-35: “They considered [,] however [,]that women would require…”

Line 65: low literacy rates[,] and immigration…

Line 66: a sentence cannot begin with a number

Line 70-71: three-year basis [not yearly]

Lines 77-79: need a comma somewhere, run-on sentence

Line 128: the dash should be a comma

In general, avoid the phrase “more disadvantaged” – saying “disadvantaged” is sufficient and clearer

Line 150-151: needs commas

Other comments:

Line 31: What are “semi-directive” interviews? Do you mean semi-structured?

Lines 46-47: these must be estimated numbers, explain and cite: “[Source] estimates that there were 604,127 cases and 341,931 deaths…..”

Line 49-54: unnecessary, the rates in France are not relevant as the inequality issue is mentioned below. Better mention global rates of CC in low-income settings and then go straight to the second paragraph in Line 55.

Lines 68-73: unnecessary quotes and detail, needs to be summarized – why is this relevant?

Lines 81-87: should go earlier in the self-sampling discussion, out of place here

Lines 88-98: unnecessary and confusing, since this manuscript is not about the trial. Instead, cite the paper or protocol of the trial when it is first mentioned and go directly to next paragraph.

Line 146-158: this is all a bit unclear, needs to be rephrased – how and why was this index used in the study? What is IRIS and what is INSEE? Why is EDI mentioned?

Lines 166-17: so a modified index was used to create an interview guide? Again, this is the main data collection procedure, so this whole section needs its own paragraph without unnecessary acronyms. The relevant index needs to be mentioned clearly and it needs to be explained how it was adapted/used to create interview guides, if that’s what it was used for.

Line 177-179: is there a source of this that needs to be cited?

195: who were the anthropologists? Part of the team? Research assistants?

95-196: how the interviews were developed needs to be explained earlier (wasn’t this related to the index?) and in more detail – where did questions come from, from the literature? Previous research? Etc.

Line 203-205: how were themes identified? Was a coding scheme created? If so, how (was this an inductive or deductive coding process)? Was there agreement among coders? How many themes emerged? Etc.

Lines 224-225, 230-231, 252-254: already mentioned, sounds redundant. These are also results, not methods, so this is out of place.

Lines 233-240: this is a confusing paragraph, participants’ profiles should be presented as table clarity as in Table 1

Lines 374-276: fragment, revise

The map legend is confusing: what is PMI? Iris boundary? (should it be IRIS?) What do the numbers under Indexes mean? Activity and others?

Fig. 3: “Family culture” is an odd term, some of the fields refer to religion and others to ethnicity and some to both. This needs to be consistent, perhaps use Self-reported ethnicity? Religion?

Fig. 4: Categories on X axis need to be capitalized

Finally, the recommendation and conclusion sections could be integrated into one, otherwise the conclusion is redundant and repetitive.

Reviewer 3 Report

The authors have satisfactorily addressed the concerns raised in the previous revision cycle. They have made significant changes to the manuscript, and have made the intent of the study sufficiently clear this time around. They have also included tables and figures to make data presentation easy to understand and analyse. The "Recommendations" section is specially appreciated, as it adds value to the aims and intent of this study. 
